# Community-Level Experiences, Understandings, and Responses to COVID-19 in Low- and Middle-Income Countries: A Systematic Review of Qualitative and Ethnographic Studies

**DOI:** 10.3390/ijerph182212063

**Published:** 2021-11-17

**Authors:** Christopher B. Raymond, Paul R. Ward

**Affiliations:** College of Medicine and Public Health, Flinders University, Adelaide, SA 5042, Australia; paul.ward@flinders.edu.au

**Keywords:** COVID-19, SARS-CoV-2, community ethnography, pandemic social science, qualitative, resilience, vulnerability, uncertainty, risk perceptions

## Abstract

(1) Background: COVID-19 disruptions offer researchers insight into how pandemics are at once biological and social threats, as communities struggle to construct meaning from novel challenges to their ontological status quo. Multiple epistemes, in which public health imperatives confront and negotiate locally derived knowledge and traditions, vie for legitimacy and agency, resulting in new cultural forms. (2) Methods: To investigate the context and construction of community responses, a systematic review of qualitative literature was conducted with the aim of evaluating those insights provided by empirical, social field research in low- and middle-income countries since the onset of COVID-19. Six scholarly databases were searched for empirical, qualitative, field-based, or participatory research that was published in peer-reviewed journals between December 2019 and August 2021. (3) Results: Twenty-five studies were selected for data extraction, following critical appraisal for methodological rigor by two independent reviewers, and were then analyzed thematically. Faced with unprecedented social ruptures, restrictions in social and physical mobility, and ever-looming uncertainties of infection, financial insecurity, stigma, and loss, communities worldwide reacted in multiple and complex ways. Pervasive misinformation and fear of social rejection resulted in noncompliance with pandemic sanctions, resistance, and increased isolation, allowing the spread of the disease. The meaning of, and understandings about, COVID-19 were constructed using traditional, religious, and biomedical epistemologies, which were occasionally in conflict with each other. Innovations and adaptations, through syntheses of traditional and biomedical discourses and practice, illustrated community resilience and provided models for successful engagement to improve public health outcomes. (4) Conclusion: Local context and community engagement were indispensable considerations when enacting effective public health interventions to meet the challenges of the pandemic.

## 1. Introduction

Global COVID-19 disruptions offer researchers insight into how pandemics are at once biological and social threats, as communities struggle to construct meaning from novel challenges to their ontological status quo [1,2]. The threat of contagion, and human efforts to contain, avoid, and eliminate COVID-19, has dominated biomedical discourse since early 2020, resulting in governments and international actors adopting technocratic, materialistic COVID-19 mitigation policies focused on isolation, physical distancing, quarantine, testing, tracing, and THE dissemination of risk communications [3,4,5]. The entanglement of the SARS-CoV-2 virus with the kaleidoscope of local socio-cultural contexts and individual behaviors confronts and confounds, while the unprecedented pandemic challenges public health systems without exception. The evolving COVID-19 crisis has exposed vulnerabilities across political, economic, health and other domains, revealing contours of inequity and differential access to information, services, power imbalances and constraints on human agency, and, according to official estimates as of 30 September 2021, has infected over 230 million people and has left almost five million dead since the pandemic erupted [6,7,8]. Unprecedented in scope and scale, COVID-19 has destabilized and reorganized how we enact relationships and use our bodies, dramatically influencing how we move, interact, and understand our place in the world [9].

Vulnerabilities affecting many nations’ pandemic response center on the confluence of limited human, commodity, and financial resources, the historical contexts influencing the management and dissemination of reliable, competent data and information, and each government’s engagement with its citizens [10,11]. As SARS-CoV-2 spread throughout low- and middle-income countries of the Global South (LMICs, defined by the World Bank and OECD as having a GNI per capita of USD 1046–4095 (https://datatopics.worldbank.org/world-development-indicators/the-world-by-income-and-region.html; Accessed on: 15 September 2021 https://datahelpdesk.worldbank.org/knowledgebase/articles/906519-world-bank-country-and-lending-groups; Accessed on: 15 September 2021)), the virus held fast amid fractured health systems and ill-prepared governments, resulting in the world’s largest share of excess COVID-19 deaths compared with other income-group countries [12,13]. Lack of access to reliable data in many countries, due to challenges in health information systems, testing and outreach constraints, and other infrastructural and behavioral issues, has forced health authorities to estimate without proper evidence, which has the double disadvantage of wasting limited resources and engendering mistrust in an already-suspicious public. Revisions in modeling estimates demonstrate that COVID-19 deaths are likely three times higher than what is officially reported to the WHO and country surveillance systems, inevitably causing protracted pandemic suffering by underestimating the needed resources to quell the crisis [13,14,15,16]. Chronically overextended health systems in LMICs have buckled under the weight of COVID-19, and pervasive rifts and ruptures in social life will likely have consequences outlasting the acute, liminal phase of this pandemic [17].

Social responses to COVID-19 are both expected and novel, as many LMICs confront structural inequality, histories of colonialism, racism, and poverty, and the legacies of recent past outbreaks and high endemic disease burdens. Almost without exception, communities have faced episodes of stigma and social rejection, panic, uncertainty, and novel negotiation of risk perception and evolving explanatory models to understand and respond to COVID-19 [18,19,20]. Many of these resource-constrained communities are already burdened by endemic infectious or non-communicable disease, as well as heavy burdens of disability. However, each community is embedded in its own history, socio-cultural milieu of language, ethnicity, religion, power dynamics, and modernity, and is unique both in its phenomenological experience of the pandemic and to what degree these experiences have destabilized social life. While instrumental in understanding best practices and approaches, prior histories of outbreaks of SARS, Ebola, and global pandemics of HIV and TB, for example, have not sufficiently prepared communities with robust systems of response and containment, both materially and socially [21,22,23,24,25,26,27,28].

Therefore, it is vital that social analyses be prioritized as policymakers and community members navigate the ever-shifting landscape of the crisis, as there are no easy one-size-fits-all approaches or policies that are effective for the vastly unique communities encumbered by COVID-19 [29,30,31,32]. As illustrated in this literature review of studies conducted around the world in LMICs, socio-cultural contexts and the minutiae of lived experience are paramount for any effective engagement or development of appropriate and participatory public health policy and intervention. As researchers, it is vital to strengthen the links between community engagement and policy development, bringing to light the details of daily life and the real contours of social suffering as a route toward the amelioration of this massive global crisis. Real change happens in the community, and globally derived discourses and policies are ineffective if not grounded in local context and understanding.

## 2. Materials and Methods

The focus of this review is to identify what is currently known about how communities in LMICs are responding to the pandemic, the socio-cultural context of each selected LMIC, and the theories brought to bear, especially as they pursue a critical inquiry into social processes and public health design. Issues of power are also under consideration, identifying how hegemonic knowledge is used, and the use of qualitative methodologies in field data collection. The fundamental research question for this review is, what do we know about the meaning of COVID-19 in the communities it is most affecting, and how is this represented in the empirical literature? Unless we understand how impacted communities frame and understand COVID-19, there is a significant risk that policies and interventions will be acontextual and will have limited impact. 

Since early 2020, government-mandated restrictions on movement, intended to reduce the potential transmission of the SARS-CoV-2 virus that causes COVID-19, have substantially disrupted field-focused empirical data collection and research methodologies. Suspension of in-person data collection at various times during the global pandemic lockdowns has spurred qualitative researchers to consider “how can qualitative inquiry, founded on human connection, empathetic listening, and ‘thick description’ advance in a world of social distancing?” [33] (p. 1061). Alternative methods were quickly proposed and adopted by academics, especially those employing qualitative methodologies, including conducting digital ethnographies, shifting to phone and online surveys, interviewing, and creating focus groups using digital platforms such as Zoom, WhatsApp, and others [34,35,36,37,38,39,40]. Several scholars have proposed research agendas to collate revisions in research topics, data collection methods, and modes of dissemination stemming from pandemic restrictions and ruptures in the status quo of data collection [41,42,43]. Epistemological debates have emerged regarding questions of data quality, contextual richness, and study rigor in the context of shifting field methods, in which concern over the health and safety of researchers and participants, in addition to ethical imperatives to ensure compliance with local and international pandemic regulations and the principle of non-maleficence, have been brought to the fore [33,37,39,44].

As qualitative research is, by definition, a “situated activity that locates the observer in the world … and involves an interpretive, naturalistic approach” [45], it follows that traditional methods of inquiry, including participant observation, in-depth interviews, and social immersion, provide the richest contextual materials with which to develop the “thick descriptions” necessary for interpretive or critical analyses of social phenomena. For this review, the authors elected to focus on empirical social research conducted at the community level, using in-person, on-site data collection to capture rich, detailed, and contextual information on the pandemic.

Inclusion and exclusion criteria were formulated for the selection of published qualitative research, available from six scholarly databases. Studies were confined to literature from LMICs experiencing similar pandemic impacts on poverty, stigma, infrastructure and supply limitations, communications, or information access barriers. Regarding the selected study methodologies, data collection methods and research approach paradigms were considered along with an understanding of the impact that global social restrictions have had on conducting qualitative field research, specifically those that engage at the community level and that explore the meanings and experiences of the pandemic.

The review focuses on empirical, primary, qualitative research conducted at field sites, and excludes studies that are not heavily reliant on face-to-face interactions. The following criteria were used for the literature review of empirical qualitative studies on COVID-19 in LMIC communities (see Table 1).

Studies published in peer-reviewed journal articles were selected following the screening and in-depth evaluation of articles resulting from database searches using the following Boolean search terms: “COVID*” AND “ethnograph*” OR “anthropology” OR “qualitative”, and variations that included terms such as “case study”, “phenomenology”, “lived experience”, “meaning”, “study”, “research”, and “empirical”. These terms were used to ensure broad capture of the available literature. Databases used for this review were Anthrosource, Google Scholar, Ovid, Pubmed, Proquest Social Science, Scopus, and Web of Science.

A total of 2152 records from seven scholarly databases were identified for initial title and abstract screening. 2067 studies were excluded, based on title and abstract appraisal as per the inclusion and exclusion criteria parameters. The remaining 85 studies were downloaded and thoroughly screened by two reviewers working independently, using the JBI critical appraisal tools for qualitative research [46] to identify those studies eligible for inclusion. Following this critical appraisal, 59 studies were excluded due to low methodological rigor, geographic ineligibility, or an ineligible study focus area. A final set of 26 studies that met the inclusion criteria and passed critical appraisal were included for review. A data extraction table was developed (see Table 2) to highlight the key domains across all the included studies. A PRISMA flow diagram of the study selection process is included in Figure 1 [47].

The selected studies were reviewed by first conducting a full reading of each one, followed by data extraction and inductive thematic analysis in which the unifying common themes were identified across all papers. Despite the diversity of topic areas in the literature, several cross-cutting themes emerged from the 26 papers, including issues concerning knowledge and information, the psychosocial impacts of the pandemic, the effects of social and mobility restrictions, investigations into governance and health system challenges, and community cohesion and innovation. These themes shared traits in common and were grouped into two broad domains of response categories: reactions and adaptations.

The notions of “reaction” and “adaptation” were conceptually useful as a means of classifying the literature included in this review, despite definitional associations with positivistic reductionism. In chemistry and physics, reactions are imbued with movement and response, occasionally in violent opposition to a disruptive force “exerted in opposition to the impact or pressure of another body; a force equal and opposite to the force giving rise to it” [48]. Implicit in this concept of reaction is the framing of novelty and of response to an unanticipated perturbation, being ontologically concordant with complexity, chaos, and dynamism. Evidence of this is clear from the social and behavioral reactions outlined in the research, in which compensation took the form of stigmatizing discrimination, alterations in health-seeking behavior, rifts and breakages in sociality, and material disruptions to everyday life. This review identified several sub-themes by which to categorize these studies, focusing on knowledge and misinformation and its effects, social and psychological experiences, impacts on mobility and social restrictions, and challenges in public sector governance and health systems.

Adaptation may be thought of as a reaction to reaction, as the cultural system seeks to level out stability and replacement from outbreak disruption and displacement. Biologically, adaptation can be considered as the process whereby an organism or species becomes better suited to its environment [49]. Sociologically, this concept is consonant with resilience, in which the attributes of a complex social system determine suitability to a new environment, in this case, an endemic or post-pandemic world [50,51]. As a counterpoint to the sub-section on reactions, two sub-themes are included that outline adaptive responses, including a focus on community cohesion and innovation incorporating local knowledge, and examples of successful public-sector governance. Adaptation, as a component of resilience, informs models of best practice that help us to better understand the dynamics of how people are adapting to jolts in the system (e.g., a global public health crisis) [52,53]. Meyer suggests that adaptations to jolts fall into three phases: anticipatory, responsive, and readjusting [53]. Most of the cases in the included literature examine the effects of lacking anticipation, focusing on responses (reactions) and readjustments (adaptations). Weick and Sutcliffe observed that “unexpected events [such as a global pandemic] often audit our resilience” (parenthetical text added by the authors) [54]. Resilience, as opposed to time-bound reaction, is a process rather than an outcome. If we consider the web of conceptions around risk and resilience, reactions, and adaptation, we form a picture of how to approach the studies included in this review from both processual and static perspectives.

While the studies investigating pandemic experiences tended towards the descriptive, relating discrete phenomena of reported reactions that were both internal and external, a more holistic framework using risk and resilience could potentially capture more of the socially dynamic and temporal aspects of the pandemic. As Evans-Pritchard discovered during his classic ethnographic explorations of witchcraft among the Azande in 1937, the site and timing of misfortune require two sets of explanations: how and why this happens, and how and why this happens to this person at this time [55]. As the respondents in these studies recalled their personal travails with COVID-19, the “what” of description begs the question of “why?”. As Panter-Brick suggested, Evans-Pritchard’s explanations of misfortune can be rephrased when posed by individual subjects in an experiential context as, “Why is this happening to me, at this particular time?” [56]. In our reading of the included literature, an important distinction can be made between the snapshot narratives of discrete experience by respondents and the need for further, more comprehensive contextualization, including answering the question of “Why these people, and why now?”.

**Table 2 ijerph-18-12063-t002:** Twenty-five community-based studies included in the literature review.

Citation	Country	COVID-19 Focus	Target Pop.	Article Title and Key Outcomes	Approaches
Adom et al. [57]	Ghana	Stigma and mental distress	HCWs and patients	The psychological distress and mental health disorders from COVID-19 stigmatization in Ghana—Stigma and psychological distress among HCWs, patients, and others; psychosocial recommendations for policy change	Phenomenology
Ali [58]	Pakistan	COVID-19 burials	Local community	Rituals of containment: many pandemics, body politics, and social dramas during COVID-19 in Pakistan—Ethnography of funeral rites in the context of COVID-19 government restrictions; changes in burial traditions; social consequences; entanglement of science, religion and politics	Social drama, symbolic ownership of the “viral body” by the state, liminality and grief, death traditions
Ali et al. [59]	Pakistan	Mental health, perceptions	Local community	When COVID-19 enters in a community setting: an exploratory qualitative study of community perspectives on COVID-19 affecting mental well-being—Anxiety and fear, social, financial and religious crises and distress. Coping: becoming closer to God and family, participating in mental health sessions, and resetting lives	Qualitative, descriptive
Amir [60]	Uganda	Stigma and mental distress	Recovered COVID-19 patients	COVID-19 and its related stigma: A qualitative study among survivors in Kampala, Uganda—Narratives of stigma experiences, social rejection, labeling and distress	Qualitative, descriptive, narrative
Asiimwe et al. [61]	Ghana	Perceptions of contact tracing	Contact tracers, contacts, and supervisors	Stakeholders’ perspective of, and experience with, contact tracing for COVID-19 in Ghana: A qualitative study among contact tracers, supervisors, and contacts—Perceptions of utility and effectiveness of COVID-19 contact tracing among implementing bureaucrats and recipients; generally positive experiences and expressed concerns of stigma associated with home visits	Phenomenology, narrative, Lipsky’s street-level bureaucrats theory
Bahagia et al. [62]	Indonesia	Local wisdom, food security and livelihoods	Community leaders	Local wisdom to overcome the COVID-19 pandemic of Urug and Cipatat Kolot societies in Bogor, West Java, Indonesia—Food redistribution, collective action through *nujuh bulanan*, instigating taboos, Indigenous knowledge that combats “life perturbations”	Qualitative, ethnography of local knowledge (ceremonies, taboos, rituals), descriptive
Bhatt et al. [63]	Nepal	Perceptions, understanding, and prevention	Local community	Perceptions and experiences of the public regarding the COVID-19 pandemic in Nepal: a qualitative study using phenomenological analysis—Knowledge measures, social isolation, inadequate PPE, disorganized public sector	Phenomenology, lived experience
Ekoh et al. [64]	Nigeria	Effects of social restrictions	Above 60-aged community	Digital and physical social exclusion of older people in rural Nigeria in the time of COVID-19—The elderly are digitally and socially excluded due to pandemic restrictions, leading to loneliness and lack of coping	Qualitative, descriptive
Ghani and Sitohang [65]	Indonesia	Knowledge and responses of community	Remote Indigenous community	How people in remote areas react to the COVID-19 pandemic in the early phase—Hoaxes predominate and circulate widely; with limited access to reliable information, there is a need to improve access to reliable information and quell hoaxes	Digital vicious cycle, “illusory truth effect”, bullet theory of communication
Jones [66]	Sierra Leone	Experiences of state-led COVID-19 measures	Urban and rural communities	An ethnographic examination of people’s reactions to state-led COVID-19 measures in Sierra Leone—Adaptation, non-compliance, passive, and active resistance; heterogeneous responses by communities	Adaptive capacity, compliance, passive, active resistance theories; social and financial capital
Kumari et al. [67]	India	Psychosocial functioning	Peripartum women	Impact of COVID-19 on psychosocial functioning of peripartum women: a qualitative study comprising focus group discussions and in-depth interviews—Peripartum women experienced distress, anxiety due to pandemic confinement, and social restrictions during and after pregnancy	Qualitative, descriptive
Kwaghe et al. [68]	Nigeria	Stigma, trauma	Frontline HCWs	Stigmatization, psychological and emotional trauma among frontline health care workers treated for COVID-19 in Lagos State, Nigeria: a qualitative study—Knowledge assessed for biomedical understanding; experienced stigma and social reactions from family and community; insights into improving health care quality based on experiences	Colaizzi’s phenomenological method
Newton et al. [69]	Ghana	Health-seeking behavior	Above 60-aged community	Understanding older adults’ functioning and health-seeking behavior during the COVID-19 pandemic in Ghana—Reporting physical and emotional health during the pandemic; challenges of loneliness and health-seeking restrictions and health provider attitudes	Qualitative Thematic Analysis, descriptive
Nicoletti et al. [70]	Bolivia	Patient experiences	Rural patients with epilepsy	The impact of COVID-19 pandemic on frail health systems of low- and middle-income countries: The case of epilepsy in the rural areas of the Bolivian Chaco—Patients with epilepsy in remote Bolivia experienced drug stockouts and lack of access to health care; 75% had inconsistent medication use during COVID-19 lockdowns	Qualitative, descriptive
Okediran et al. [71]	Nigeria	Experiences and perceptions	Frontline HCWs	The experiences of healthcare workers during the COVID-19 crisis in Lagos, Nigeria: a qualitative study—Four themes identified around responsibilities, challenges and coping strategies, experiences of distress and pleasure, and recommended needs for further material and social support	Qualitative, descriptive
Østebø et al. [72]	Ethiopia	Religious and secular perspectives	Local community	Religion and the “secular shadow”: responses to COVID-19 in Ethiopia—Conflations of science and religion, tradition and modernity in the Ethiopian context as local perceptions are considered in the development of public health interventions, exploring epistemic tensions	Qualitative, ethnographic, Latour, coexisting epistemologies, modernity
Prajitha et al. [73]	India	Government responses	Government bureaucrats	Strategies and challenges in Kerala’s response to the initial phase of COVID-19 pandemic: a qualitative descriptive study—Five themes emerged in reflecting on government responses, recognizing key components of social capital, a robust public health system, participation and volunteerism, health system preparedness, and challenges	Qualitative, descriptive, social capital, SDH
Prasetyo et al. [74]	Indonesia	Civil society participation	Task Force members	Civil Society participation in efforts to prevent the spread of COVID-19—Four task forces engaged: public education, controlling mobility via gate system, hand washing, and food needs/suspected patient monitoring, etc. Lack of funds and lack of public awareness were the main obstacles	Civil society engagement
Prasetyono et al. [75]	Indonesia	Leadership and local governance	Village heads	Patron-client relationship between village heads and their residents during the COVID-19 pandemic—Village leaders influence public opinion and awareness, consolidate volunteers and information, facilitate social assistance. Patron-client relationship between village head and residents, seen as a “father protector”	Qualitative, patron–client theories, power relations in bureaucracy
Samuelsen and Toé [76]	Burkina Faso	Ruptures in politics and life	Local community	COVID-19 temporalities: Ruptures of everyday life in urban Burkina Faso: Investigated community responses to government-led restrictions as prevention prior to the advent of COVID-19 in Burkina Faso, placed within the socio-economic, political, and fragile security contexts at the time	Qualitative, anthropology, Giddens “time-space distanciation”, outbreak narratives
Sari et al. [77]	Indonesia	Social protection with village fund	Community and leaders	The Effectiveness of Tri Hita Karana-based traditional village management in COVID-19 prevention in Bali—Experience in managing the village fund for social protection during COVID-19, using traditional *Tri Hita Karana* philosophy. Local wisdom provides positive outcomes for village resource distribution and social protection	Tri Hita Karana Hindu philosophy
Sharma et al. [78]	India	Information, media, andpsychosocial experiences	Local community	Panic during COVID-19 pandemic! A qualitative investigation into the psychosocial experiences of a sample of Indian people—Misinformation causes panic and anxiety; quarantines and social restrictions created cognitive dissonance	Qualitative, descriptive, social psychology, grounded theory
Sukmawan [79]	Indonesia	Traditional rituals	Local community	Tradition-responsive approach as a non-medical treatment in mitigating the COVID-19 pandemic in Tengger, East Java, Indonesia—*Nambak lelakon*, an adaptation of traditional *tolak bala* ritual in East Java, used to maintain and protect human life through the collective non-medical mitigation of COVID-19. Use of this ritual instills harmony in the community and is a form of prayer and surrender to God	Qualitative, psychosocial, religious
Sumesh and Gogoi [80]	India	Stigma, discrimination	Recovered COVID-19 patients	Collecting the “Thick Descriptions”: A pandemic ethnography of the lived experiences of COVID-19-induced stigma and social discrimination in India—Embodied experience of stigma; former patients discriminated against and criminalized; social process of stigma analyzed	Pandemic ethnography, lived experience, grounded theory, Geertz, Goffman, narrative
Tan and Lasco [81]	Philippines	Local knowledge	Traditional community	‘Hawa’ and ‘resistensiya’: local health knowledge and the COVID-19 pandemic in the Philippines—Ethnographic study of “contagion” and “immunity” framing in illness understanding and explanatory models for COVID-19; multiple ontologies/traditional knowledge	Ethnography, postcolonialism, risk theory, political economy
Wibisono et al. [82]	Indonesia	Religious exclusion and xenophobia	Muslim community	Turning religion from cause to reducer of panic during the COVID-19 pandemic—Explored ways to reduce social exclusion and reactions via religious cohesion in a traditional community	Collaborative auto-ethnography, Weber’s verstehen, Geertz

## 3. Results

The following sub-sections outline the results of ethnographic investigations undertaken since the advent of the COVID-19 global crisis. Anthropological research into local explanations and experiences of COVID-19 fosters deeper insight into the observed social impacts of the pandemic, including social exclusion and stigma, blame, panic, and mistrust [18,19,83]. Plague and pandemics are characterized by “cycles of shame and blame, stigmatizing discourses and isolation of the sick” [84] and yet offer opportunities for interrogating cultural resilience, as outlined by a few of the included studies.

The reviewed studies are grouped by broad conceptual considerations, for coherence and simplicity. As a set, the papers follow the story arc of the outbreak narrative: initial disruptions, displacement, and urgency issued by an uncertain outbreak, followed by regulatory and social (over)compensation designed to ameliorate its effects; a period of interference and instability, infection, and response; and the early appearance of re-stabilization in which local and exogenous explanations and relationships are cohered and synthesized into new cultural forms. Themes are thus grouped into responses arising from reactions to the pandemic, clustering around knowledge and misinformation, social and psychological effects, the impacts of social restrictions, and challenges in governance. Adaptations cluster around research illustrating community cohesion and adaptive governance (see Figure 2). 

### 3.1. Reactive Responses

#### 3.1.1. COVID-19 Biomedical Knowledge and Misinformation

Knowledge management during pandemics and natural disasters is a contested site of scholarly engagement, including who decides what knowledge is legitimate, usually following lines of power and authority, freighted by the influence of hegemony. The biomedical paradigm dominates the COVID-19 global discourse, thus linking scientific knowledge with legitimacy [5,85]. However, with the advent of increased access to unreliable sources of information via online or other media, communities are bombarded with competing knowledge paradigms, manifesting as hoaxes, conspiracies, misinformation, and disinformation, in addition to local ontologies that may run counter to biomedical epistemology [86,87,88,89,90].

Investigations into how pandemic information and knowledge arrive at, and are interpreted by, the community were achieved through enquiring into the perceptions and understandings of COVID-19 [59,63,65,66,68,72,76,78,81]. Information access and its effect on perception and experience were linked to education and geography, with educated, urban, younger populations having better access to technology and social media [64,65], yet the perceived or reported reliability of information sources was not described in depth. Rural, low-income populations were associated with lower literacy rates and poorer access to health information sources [65]. Access to knowledge was also seen as a primary driver of risk perceptions, as many hoaxes and sources of non-biomedical conspiracies altered risk behaviors, including the postponement of the seeking of care at health facilities, potentially worsening health outcomes [65,78].

The proliferation of rumors was cited in these studies as the principal driver for “misunderstanding” COVID-19, filling gaps in information access with fictions arising from superstition and exaggeration. Researchers in East Java, examining civil society participation in COVID-19 mitigation, noted that many villagers held the belief that “the coronavirus did not exist and that it was just a conspiracy deliberately invented in the interests of capitalists” [74]. In remote West Kalimantan, on the Indonesian island of Borneo, the Sebaruk Dayak community was reported to have a limited understanding of the COVID-19 virus or its transmission. An incident from one village quickly circulated on social media via WhatsApp, effectively muting and replacing biomedical information: 

*Believe it or not. This afternoon, did anyone hear the thunder when it was hot (not raining)? There was a true incident from Popay today. A baby was born this afternoon, and before the attendants could cut the umbilical cord, it spoke and said, “To avoid the coronavirus, you must boil an egg”. Immediately after, a strong thunderclap was heard and the baby began crying* [65].

The researchers reported widespread dissemination of this rumor, which had a negative impact on people’s understandings and behaviors. They ascribed the lack of acuity in accessing information in the region to “a lack of cognitive ability and experience with information technology” [65]. 

In Sierra Leone, rumors of police bribes by the wealthy to enable free movement during social lockdowns revealed how suspicion arises from economic inequality. Individual subjugation to the regulations of the state was perceived as uneven, with preference given to the privileged in society [66]. The circulation of these rumors, even assuming some basis in truth, had a negative impact on the management of pandemic information and, thereby, on community responses.

Two important components of information access were outlined by these studies, including misinformation and differential access to reliable information. Sharma et al., referencing “misinfodemics”, noted that the proliferation of misinformation circulating among Indians “worsened the impact of the pathogen, and caused agitation and frustration” among respondents [78]. Media-driven misinformation was amplified through collective xenophobia, as perceptions of COVID-19 carried by outsiders was a common theme, both in religious institutions in Indonesia and among the general population in India [78,82]. In Burkina Faso, despite the mass media focus on the Chinese origins of the pandemic, communities located the viral genesis squarely in Europe (France), referencing the “disease of the whites” associated with European wealth, thus distorting the understanding of the temporality and locality of the pandemic, with assignations distinct and separate from previous infectious disease outbreaks, such as Ebola [76]. Similarly, Ethiopians initially viewed the pandemic as a “white man’s disease which struck the Western world due to immorality and sin” before the first COVID-19 case arrived [72]. It then transformed into a local affliction, God’s punishment for people’s sins and transgressions, triggered by a multiplicity of wrongdoing [72]. 

Limited access to “reliable” information was directly linked by the authorities to poor compliance, assuming a lack of understanding and general awareness of COVID-19. In contrast, excess access to disinformation and misinformation was reported to drive stigmatizing and ostracizing social behaviors among healthcare workers, in which the fear of transmission overrode protective behaviors established by health facilities [68,71].

Observations from the studies included in this review align with the widely reported circulation of misinformation around the world pertaining to COVID-19 [91,92,93,94,95,96]. As risk and public health communication is fundamental to any containment or mitigation strategy, the empirical data on constraints to information, as well as the proliferation of misinformation, has had a demonstrable negative impact on the overall handling of the pandemic [97,98,99]. However, apart from Tan and Lasco’s study on local knowledge in the Philippines [81], Samuelsen and Toé’s work in Burkina Faso [76], and Ali’s ethnography of death in Pakistan, the reviewed literature shows little consideration of the tensions around how, and which, knowledge came to be legitimized, by whom and for whom it was circulated, and how to best anticipate and incorporate strategies adapted to local epistemologies and explanatory models. This is an important notion going forward, in the context of the social construction of a pandemic.

#### 3.1.2. Social and Psychological Effects

Most studies in this review observed the psychosocial and economic impacts of the COVID-19 pandemic in LMIC communities around the world [57,58,59,60,63,64,66,67,68,71,78,80]. The public health literature on pandemics and epidemics of infectious origin is rife with accounts of how these catastrophic disruptions to ordinary life are met by disarray and uncertainty, spawning fear and heightened anxiety [19,20] and often resulting in marginalization and stigma, as communities grapple with unknowns and avoid risks [8,100,101,102,103,104]. Inevitably, these rifts in the social fabric can have profound psychological and sociocultural effects, such as emotional distress, economic exclusion and loss of livelihood, and the fostering of mistrust and suspicion; they can also engender stigmas resulting in social rejection [103,104,105,106,107]. 

The psychological and other social impacts of disease outbreaks can be understood as downstream proximate effects that often have far-reaching upstream causes beyond the merely interpersonal or the immediately apparent [108]. Individual experience and perception are embedded in a matrix consisting of sociocultural context and the threads of a number of structures and systems, including inequities arising from the global capitalist economy, political and racial inequality, violence, and other causes [56,107,109,110,111]. 

Feelings of rejection and isolation resulting from the stigmas attached to COVID-19 uncertainties were reported among healthcare workers, patients, caregivers, and people at high risk of infection. Kwaghe et al. and Okediran et al., in two separate studies on healthcare workers and stigma in Nigeria, described exhaustion and fear arising from concerns about transmitting the virus to family members, overwork from added duties, and from community rejection as suspicious, viral entities [68,71]. They also avoided providing comprehensive services out of the fear of catching COVID-19 while engaged in their duties, leading to feelings of guilt and frustration. In Nepal, Bhatt et al. provided evidence of extreme social stigmas from COVID-19, including patients being wholly shunned by the community and not being allowed to return home, despite negative PCR results, and healthcare workers being ejected from housing [63]. Health workers in Nigeria and Ghana were subjected to a spectrum of stigmas, suspicions, and ostracism, including by their colleagues, after being treated for workplace-acquired COVID-19 infections [57,68,71]. Certain religious groups were also blamed for the spread of COVID-19, and local government officials and the police responded to suspected cases with violence and the destruction of property [63]. 

The physical quarantining and isolation of suspected or confirmed patients as a means of reducing transmission and mitigating COVID-19′s effects in the community had the unfortunate side effect of generating stigma, discrimination, labeling, and social rejection from the community and families, and on occasion, from healthcare providers [57,60,80]. Actual and symbolic labeling of the houses of quarantined patients, regardless of diagnosis, resulted in protracted avoidance and stigma. In one study in India, local health departments affixed large red banners outside quarantined households, admonishing passers-by to avoid the area, effectively delineating the space as unclean and labeling it as a “containment home” [80]. Unlike other pandemics of periodic or routine incidence, such as cholera or dengue, the unknown and ontologically insecure place of COVID-19 instills a fear and reflex of rejection when confronted with symbolic labels, such as “containment home”, that inevitably result in social ostracization. Ugandans and Ghanaians, still smarting from the relatively recent stigmatizing outbreaks of SARS and Ebola [102,112], also experienced taunting, shunning, and social rejection through labeling practices [57,60]. “Corona families” were unable to shop in their usual markets, being thrown out or physically excluded by vendors [60]. In Ghana, vendors reported being banned from selling products in local markets, resulting in both psychological and economic consequences [57]. Respondents in these studies reported a profound sense of loss, isolation, sadness, and depression arising from the lived experience of stigma and social rejection. 

Sumesh and Gogoi, in their “pandemic ethnography”, drew comparisons between the phenomenological sense of bodily materiality and restriction as the site of pandemic control and the historic construction of “untouchability” associated with the Indian *Dalit* caste [80]. In this sense, the stigma and rejection associated with quarantine and the labeling of “unclean” COVID-19 patients were likened to the structural inequality arising from caste members labeled as “untouchable”. The imperative of containment of contagion, whether biological or social, takes advantage of the mechanism of stigma during acute times of instability, using it as a means of reducing uncertainty and panic among uninfected, untainted, or “morally pure” subjects [103,105,113]. 

Mental distress took many forms in these studies, including worry and concern about the future, anxiety about finances, guilt, depression, grief, fear, loneliness, and mistrust. The fear of contagion and death figured prominently in these studies, made more pronounced by social isolation and exaggerated media reports that spurred anxieties among the respondents [57,58,60,63,67,68,78,80]. Fear of the future was illustrated by a spectrum of concerns, as reported by researchers in Pakistan, centered on anxieties about the pandemic’s effects on children, the nature of the post-COVID-19 world, and how many deaths would scar the future [59]. New mothers were particularly anxious and fearful of the risk of infection in themselves or their newborns, and the concomitant stigma that would likely follow [67]. 

Fear was also voiced by patients and community members who expressed worry about hospitals as centers of contagion, leading to the avoidance of health facilities and the altering of routine health-seeking behaviors [67,69]. This phenomenon exacerbated health service inequities that were tied to limited access to information, the circulation of rumors, poverty and constraints on personal agency, and increased vulnerability for non-COVID-19 patients with routine or chronic disease [59,69]. Peripartum women in India were particularly at risk for pregnancy-related complications, due to reductions in obstetrics services and anxiety over the fear of infection from health facilities [67]. Patients also feared the consequences of a COVID-19 diagnosis and were reluctant to reveal whether they had, or were suspected of having had, the disease [68]. Older adults, often confined or isolated, experienced disruptions in routine health-seeking, especially given their limitations of access to communication or digital technologies [69]. The biosocial nature of COVID-19 as a “syndemic” has profound effects on the health system burden, adding to the personal overwhelm and desperation of patients confronting multiple illnesses, economic catastrophe, and intense emotional distress [114].

Grief and feelings of loss accompanied alterations in tradition, particularly around restrictions in burial practices associated with deceased COVID-19 victims. Inayat Ali described a sense of cultural transience in the way that the state has transformed deceased family members into “viral bodies” to be controlled and regulated, negating the community’s need for closure and ritual as the dead depart [58]. Death, paralleling COVID-19 itself, is a complex socio-cultural, economic, and political event, beyond simply a biological cessation. Presenting an ethnographic account of death in the Sindh province of Pakistan, Ali contrasted the traditions of the *Namaz-e-Janaza* prayers of Islamic death rituals with the disruptive insertion of state control due to COVID-19 mitigation [115]. Stability in the outward ceremonial forms of worshipers’ relationship with Allah via the liminal space of death had been substantially fractured by COVID-19, bringing deep loss and suffering to the community. He refers to the liminal “betwixt and between” aspect of traditional death rituals, which are crucial for the mourning process, being commandeered and controlled by the state during the time of COVID-19 [115]. State regulations are divisive, in that rural, under-resourced communities are unable to fully comply with them, adding to the stress of mourning.

Emotions of stress, panic, and anxiety associated with the loss of livelihood, especially in resource-poor communities, were reported by several authors [63,76,78,81]. The loss of a sense of personal agency and control, concerns over the ability to provide future stability for families, and a looming feeling of isolation and being locked in resulting from job loss and stay-at-home orders were universally reported by the studies conducted in Burkina Faso, India, Nepal, and the Philippines [63,76,78,81]. The psychological ramifications of job loss were linked with food insecurity, a lack of ability to care for parents, having to quit school due to financial restrictions, and the general economic pressures of daily survival. The depletion of savings, insufficient funds for health care, and having to choose between eating and medicine were mentioned by respondents in several studies [57,65,81]. By contrast, government officials in Nepal were seen as escaping COVID-19-induced economic devastation relatively unscathed, due to their stable salaries and their ability to work from home [63]. Generally, there were strong associations between poverty and perceived risk, contributing to the statements of frustration, fear, and anxiety associated with job and income insecurities.

#### 3.1.3. Impacts of Social and Mobility Restrictions

Lockdowns, physical distancing, masking, hand washing requirements, restrictions in business hours of operation, and general regulation of population mobility were universally employed interventions for the containment and mitigation of the pandemic, as reported by all the selected studies. These restrictions, designed to lower viral transmission and reduce burdens on the health and financial systems, exerted far-reaching social sequelae among the public. Examples from Burkina Faso illustrated how constraints in movement and reductions in business hours had a negative impact on people’s access to economic opportunities, translating into a loss of income and financial insecurity, as previously mentioned in the above section [76]. Beyond the immediate psychological impacts of job loss, lack of economic security was associated with constraints in the ability to comply with government-regulated social restrictions and increased vulnerability [76]. Several studies observed that marginalized populations in pre-pandemic settings, such as indigenous or poor communities in all settings, were at pronounced risk during the pandemic, compounded by the ever-present possibility of natural disasters, such as earthquakes, floods, and the pandemic itself [59,65,66].

Studying state-led pandemic regulation and social restrictions, Jess Jones found that older community members in Sierra Leone were more vulnerable to increased poverty, caused in part by limitations in access to digital finance platforms and their lack of mobile phones [66]. She also outlined an analytical framework of community responses centered around adaptation, non-compliance, and active and passive resistance. Non-compliance arose from constraints in agency resulting from economic hardship, rather than from a deliberate choice to reject state regulation. She also found that resistance took many forms, from the passive resistance of taxi drivers ignoring curfew restrictions and continuing to work, to active forms of resistance, such as youth groups protesting lockdowns and storming public facilities [66].

Poverty and inequality were cited as structural barriers to complying with lockdown orders in the Philippines. Note that “*hawa*” (an immunity-contagion concept) takes on moral and social dimensions as “*microbiopolitics*”; the poor and colonized are pathologized, controlled, and locked in by the elites [81]. The infrastructural requirements of lockdowns include access to running water, soap, face masks, and proper nutrition, and yet these same life-protecting commodities are the barriers that the poor must surmount when deciding between compliance with social distancing and their livelihood [81].

In Nepal, Bhatt et al. observed that respondents in their studies were unable to strictly follow stay-at-home orders and restrictions in mobility, as low-wage-earners could not access proper “work from home” infrastructure (technology) and flexible job types, and thus were forced into the dichotomous situation of “livelihood versus compliance” [63]. Economic pressures seemed to override risk perceptions, both for breaking regulations and the increased possibility of catching COVID-19 in public areas [63,81]. Exacerbating these pressures were reports of the influence of misinformation on lockdowns resulting in job loss, reduced personal income reduction in the economy, and mistrust [78].

Among a sample of peripartum women in India, restrictions in movement and confinement to the home resulted in expected feelings of anxiety, isolation, and worry vis-à-vis the reduction in antepartum care, and concerns about their ability to care for newborns at home without the normal support of their extended family [67]. At what is usually a stressful time, pregnancy and birth were appreciably emotionally heightened in the context of the pandemic, as many new mothers were traditionally expected to relocate to their in-laws’ homes, yet they feared infection and therefore avoided this normal social behavior [67]. The prohibition of a traditional “sixth day” ritual for newborns fractured communal cohesion and the connection to heritage, causing rifts in family life and existential anxiety in the new mothers from the absence of spiritual recognition of the newly born infant [67].

Nicoletti et al. found that lockdowns and mobility restrictions in the Chaco region of Bolivia were detrimental to people with epilepsy [70]. In these remote areas, shortages of antiseizure medications, decreased availability of healthcare access at remote health centers, and the inability of patients to transit to access sites were blamed for three-quarters of epilepsy sufferers not being able to access regular medications. The issue of syndemics during COVID-19, including disability, endemic infectious and non-infectious diseases, has revealed structural weaknesses in health systems worldwide, as the diagnosis and management of diseases such as HIV, malaria, and TB have suffered during the pandemic, halting and reversing progress by up to a decade toward international Sustainable Development Goals [12,114,116,117,118,119,120]. The Bolivian study illustrates one of many concomitant health issues that have been dramatically impacted by deficiencies in the supply chain, equitable access to health care, and lockdown restrictions on vulnerable communities.

Social restrictions have reduced communal and social cohesion, especially centered around worship, ritual, temple attendance, and related religious activities. Community dependence on religion as a balm against the uncertainty and chaos of the COVID-19 crisis has led to frustration and disappointment as places of worship have been closed during the lockdown. Members of the Muslim community in Pakistan have expressed a longing to return to the mosque, and have felt incomplete and dissatisfied by their inability to come together in collective prayer [59]. As is reminiscent of Inayat Ali’s ethnography on death, Ali et al. observed that restrictions in the normal, proper burial practices, such as *ghusl*, *kafan*, and prayer, created heightened fear and anxiety in the community as their members were afraid they would die and be denied a Muslim funeral [58,59]. Limitations on collective spiritual activities were generally found to have a negative overall impact on community health and wellbeing, as reported in this literature. Wibisono et al. observed how disruptions in mosque opening in remote West Java created social insecurities manifesting as xenophobia toward outsiders and non-Muslims, which was attributed to the circulation of social media and the lack of community cohesion normally arising from the routine practice of Islamic daily prayers [82]. Elderly Nigerians, in the wake of bans on religious gatherings (weddings, births, deaths), lamented the lack of a spiritual buffer as afforded by the rituals of collective prayer: “I can’t even go to church to pray to God so that he will protect us from all these things. The whole world is paying for their sins” [64].

Østebø et al. found that Ethiopian Muslims and Orthodox Christians alike viewed the COVID-19 crisis as having a supernatural origin related to punishment for sin [72]. Remedial actions thus required fasting, prayer, and repentance to ameliorate God’s wrath. When the government forced closures of places of worship, communities felt despondent, perceiving that they had been abandoned by the authorities at a time of desperation in which the only solution was sought through communion with the divine, and only within the consecrated spaces of the church. The ineffective dissemination of restriction information and closure policies within the church also played a part in reducing confidence and disrupting social cohesion. 

Despite the demonstrable reduction in disease transmission through physical restrictions of movement, lockdowns, and confinement from a biomedical perspective, most studies revealed that the urgency with which lockdowns were instituted resulted in social shockwaves, whereby communities reeled from the effects of cultural, physical, spiritual, and familial rifts through these injunctions. Cultural transitions and the evolution of traditions are normative, yet the rapidity with which the COVID-19 pandemic forced these changes was disruptive; it will take time to establish novel adaptation and homeostasis.

#### 3.1.4. Pandemic Governance and Health System Challenges

There was a general acknowledgement of the failings in effective governance of the pandemic, including ill-prepared health systems [85]. Despite ample opportunities to develop systems and responses to potential infectious outbreaks from prior experience with H5N1, SARS, Ebola, avian influenza, and other diseases, many governments and health systems were unable to quickly adapt their infrastructure, policy, and financing to meet the challenges of COVID-19 [121,122,123,124,125]. Overwhelmed and under-resourced, health systems buckled under the increased strain of patients, a lack of PPE, and pandemic panic. Nepali respondents identified the poor implementation of tracking and tracing by the government and a lack of provision of PPE as examples of ineffective public sector responses to the pandemic [63]. Reusing surgical masks in lieu of available PPE and shifting resource allocations from chronic disease management to acute COVID-19 treatment without sufficient capacity were complaints made by patients and providers alike, illustrative of larger structural deficiencies and challenges [63]. A lack of PPE, funding shortfalls, and poor public sector coordination were identified as negative stressors on contact tracers working in Ghana, constraining public health efforts to trace potentially exposed members of the public [61]. Further exacerbating material constraints were examples of policy discordance at multiple state levels, translating into confusion and difficulties in implementing health programs on the ground [73].

Other studies reported a lack of promised or committed government assistance to communities and individuals as generating mistrust and loss of confidence in the public sector [59,73,74,76]. In remote Borneo, many health facilities were simply closed during the lockdowns, effectively reducing healthcare access among vulnerable rural Dayak communities to zero [65]. Systemic deficiencies in public-sector funding commitments for the provision of mental health services were cited in Pakistan as a key area in need of improvement for supporting community resilience [59]. Researchers in Burkina Faso noted a “contradictory relationship to the state: mistrust is rife, but more support is relentlessly sought” as communities struggled financially with the closure of markets, loss of income, and lack of government support [76].

COVID-19 operates at the intersection of the institutional and the personal, where entanglements of tradition and governance create new adaptations via the rituals of the state in managing contagion. Inayat Ali, in his ethnography of the “rituals of containment” surrounding Pakistan’s management of “viral bodies”, tackled questions of governance and who owns the dead during a pandemic [58]. He views the pandemic as having multiple dimensions: social, structural, economic, emotional, psychological, and political. His analysis of the procedures concerning deceased COVID-19 patients was seen through the lens of the state as conducting rituals of containment by commandeering bodies, isolating them from family members, and taking ownership to offset the tangential effects of further contamination.

Ali used Turner’s social drama [126] as the framing metaphor for conflict arising from tensions between tradition and governance, with the pandemic creating new subjects through the enactment of state-led rituals of containment that run counter to burial and death customs. His use of social drama provides theoretical depth to examining how the state uses soft and hard power to exert containment, which is often at odds with a critical and disagreeing public. Viral bodies, defined and managed by the state, are useful for considering the liminal quality of the pandemic, in which stable social practice is supplanted by crisis management and social upheaval. A government-sponsored national broadcast of a *Namaz Aaft* ritual (meant to reverse a curse or misfortune) was viewed as political coercion to conflate magical thinking with a scientific response, placing the blame for COVID-19 and its subsequent social dislocations in the realm of the supernatural as punishment from Allah, thereby exonerating the government and relieving them of responsibility and accountability [58]. Further dislocations erupted from the government’s containment of “dead viral bodies” through restrictions in family access to burial rituals. It was widely rumored that the wealthy were able to pay for access to conduct death rituals and prayers, while the poor were not. According to the author, the denial of last rites for the poor, while privileging the rich, illustrated the invisible hand of structural violence perpetrated through the wielding of power in Pakistan [58].

The negotiated space of tension between the structural imperatives of biosecurity instituted by public sector actors, and the social boundaries and responses of governed communities, was evident in several studies [58,63,66,76,81]. Expectations of compliance and rule-following were built into many policies, occasionally driven by misinformation, as was reported in the Philippines study [81]. To support the rollout of seemingly harsh social restrictions, Filipino bureaucrats appealed to their constituents using familiar, albeit misleading, cultural myths or conspiracies to garner support and compliance [81]. Political leaders touted unproven remedies and health supplements, and the authors critiqued this practice, noting that it was a method for absolving the authorities of responsibility in the face of a rapidly spreading pandemic [81]. 

In what Tan and Lasco refer to as the “political economy of contagion”, COVID-19′s impacts among the poor and marginalized was attributed to the state machinery, as oligarchies were largely insulated from the effects of the pandemic, while the poor were “locked out and locked in”, unable to negotiate agency against a government apparatus of control [81]. As impoverished communities bore the brunt of the consequences of lockdowns, the state navigated and straddled the imperatives of biosecurity, closing international borders and confining citizens through mobility restrictions, and the imperatives of duty of care to its citizens. Histories of terrorism and political instability, as reported from Burkina Faso, previous destabilizing outbreaks, as exemplified by Ebola in Sierra Leone, and the corrupt use of power severely restricted public-sector responses, reduced public trust, and resulted in negative health outcomes for the most vulnerable among the population [66,76,81]

In an Indian study, Sumesh et al. found that local health departments were engaged in the practice of labeling COVID-19 patients’ households as contaminated, yet did not provide social support or information to counteract the expected stigma [80]. Victims felt rejected and unsupported, lacking access to fulfill basic needs while being locked in by the government. In Sierra Leone, community resistance to public-sector sanctions arose from a sense of powerlessness and voicelessness among the public, using their bodies to protest in the absence of other social capital with which to negotiate [66]. In both these cases, structural inadequacy and inequality led to a form of forced non-compliance, negotiated through imbalanced power relations.

The LMICs profiled in the literature review suffered substantial economic and social losses during the pandemic when faced with ill-equipped health and governance systems resulting from long-standing global inequities. Although many of the countries profiled had confronted numerous natural disasters and had experienced previous infectious disease outbreaks, such as SARS, H1N1, and Ebola, the all-encompassing nature of COVID-19 had devastating effects on local economies, supply chains, social cohesion, and individual resilience. The published articles in this review were in general concord that public-sector systems failed to meet the needs of the populace and were ultimately responsible for the poor performance of government interventions.

### 3.2. Adaptive Responses

#### Community Cohesion and Adaptive Governance

Examples of resilience and responsiveness to the disruptive pandemic crisis were given by several authors, providing insight into how communities come together in times of distress and need. In Indonesia, two examples of *gotong royong* are provided in Java and Bali, promoting and encouraging local knowledge and traditional practice as a means of social protection and community cohesion [62,77]. *Gotong royong* is an Indonesian concept of mutual assistance, working together, and sharing the burdens among the community, and is an often-referenced term used particularly during natural disaster or calamity, whereby leaders invoke *gotong royong* to inspire and promote partnership and unity in times of uncertainty. The Balinese traditional philosophy of *Tri Hita Karana* was used as a framework for the redistribution of village funds to offset the economic devastation to the tourism industry, among other pandemic effects. *Tri Hita Karana* is a three-component philosophy, guiding the Balinese community (Krama): devotion to God (*Parahyangan*), kindness to fellow humans (*Pawongan*), and compassion for nature (*Palemahan*). The *Tri Hita Karana* philosophy is derived from the values of Balinese local wisdom (*Sad Kertih*) with the aim of purifying the soul (*Atma Kertih*), preserving forests (*Wana Kertih*), lakes (*Danu Kertih*), the sea and beaches (*Segara Kertih*), promoting social harmony and preserving nature (*Jagat Kertih*), and improving the quality of human resources (*Jana Kertih*) [77]. The management and disbursement of funds became a communal affair, promoting trust, transparency, and providing much-needed economic relief to families. By couching the redistribution of economic resources within a cultural wisdom frame, the participating Balinese villages have preserved their traditions while meeting the needs of an unanticipated disaster [77]. 

The redistribution of food in West Java was conducted using a complex socio-cultural methodology among the Ciptagelar, Baduy, Urug, Cipatat Kolot, and Naga indigenous communities [62]. A three-tiered process consisting of annual rice harvest distribution to vulnerable people such as the elderly and orphans, collective food preparation by household matrons for redistribution to pregnant women, through a process called *nujuh bulanan,* and instituting strict temporal taboos on shifting land ownership and farming practices, took place for the duration of the pandemic. This process prevents food insecurity, especially among the vulnerable in the communities, and the utilization of local knowledge and wisdom is seen as a key mitigator of the pandemic [62].

Two studies conducted in East Java detailed the role of local leadership and civil society involvement in successful pandemic strategies [74,75], also utilizing the traditional concept of *gotong royong*. A patron-client theory was used to approach the way that local leaders were regarded during the pandemic. The researchers found that the village heads shaped public opinion and perceptions of COVID-19, served as a consolidating center for volunteers and a conduit for information, and facilitated social assistance. They were also seen as “father protectors” of the village during the pandemic, serving in the patron-client role, as identified in the study [75]. Elsewhere in East Java, a village-level study looked at community cooperation via the involvement of civil society in the formation of four local COVID-19 task forces responsible for educating the public on health protocols, limiting population mobility, ensuring adequate infrastructure for handwashing, identifying social protection needs and food insecurity, and assisting with identifying potential COVID-19 infections in the community. The *gotong royong* concept served as an instigator for community cohesion, leadership, and effective response.

The deepening of faith and enhancing spiritual life was seen as adaptive and provided a source of hope among communities, as explicitly reported in studies from Ghana, Indonesia, and Pakistan [57,58,59,79,82]. Prayer, meditation, and ritual, when allowed, were important activities that respondents perceived as being useful in reducing anxieties and “nurturing the soul” [59]. Faith in God provided external validation and psychological support, both spiritually and as a community-building experience. A deeper analysis may conclude that the use of religion as a technology of explanation and adaptation is socially constructed in parallel with biomedical technologies, circulating together, yet with differing levels of access; in essence, they are complementary ontologies operating in tandem in the community and with varying influence on behavior and the generation of risk perception.

The Tengger tribe in East Java employed ritual technology steeped in Indonesian tradition to combat the effects of the pandemic and to promote harmony in the community [79]. The *nambak lelakon* is the Tengger’s version of the well-known and widely used *tolak bala* ritual, used to ward off a variety of misfortunes. The Tengger recognize COVID-19 as a virus; it is classified as *pageblug* within their ethnomedical system, grouped together with misfortunes such as earthquakes, tsunamis, and other natural disasters yet to come. Personified as *butha kala*, according to their cosmology, COVID-19 represents unseen negativity, and must be “sent back in the direction from which it arrived” [79]. *Butha kalas*, such as COVID-19, are appeased through ritual offerings during the *nambak lelakon*, with the aim of reducing human suffering, warding off the misfortune of the pandemic, and restoring order and harmony. The ethnographic investigation into this process demonstrated that the ritual helped the community, as the pandemic was positioned within a longstanding tradition and cosmology, with the remedy spanning both the spiritual and physical realms [79].

The supernatural coincided with the secular in pandemic practices in Ethiopia, in which a “blurring of boundaries” was seen: COVID-19 was at once a virus and yet had divine origins, being sent by God as a curse [72]. The epistemes of religion and technocratic biomedicine ran in parallel, with the identity of the disease consisting of natural and perceptual elements. This blurring of boundaries served to legitimize multiple epistemes in Ethiopia, resulting in the adoption of biomedical, traditional, and spiritual remedies to treat the illness. Østebø et al. made analytical use of etiological duality in non-Western pluralistic medical systems, as initially described by Foster [127], in which personalistic and naturalistic paradigms can co-exist. The oscillations and entanglements of prescriptive spiritual “dos” (such as prayer and fasting) and the prohibitive “don’ts” of public health regulation (such as altering social behaviors and avoiding certain practices), allowed communities to employ multiple, non-contradictory epistemologies to meet the challenges of the pandemic.

Tan and Lasco offer insights into the process of the incorporation of novel disease entities, such as COVID-19, into local lexicons of illness and politics [81]. The Filipino concepts of *hawa* (contagion) and *resistensiya* (immunity) provide insight into a biological and moral–political landscape of ideas into which COVID-19 has been inserted. *Hawa* takes on the form of both biological and social contagion, and structures local understanding as well as behavior, including the delineation of infectious physical bodies, while also ascribing moral responsibility, guilt, and accountability for contagion. *Resistensiya* establishes opposition and immunity, influences risk perceptions based on biological and cultural traits and is used as a substrate for political or moral non-adherence. The authors demonstrated how these contested concepts are negotiated in the Philippines, at once legitimizing traditional medical lexicons and setting up tensions between the community and governing structures in question [81].

Several examples of adaptive and resilient governance were identified by researchers in India and Ghana [61,73]. Thematically, successful public health interventions relied on intersectoral collaboration, building upon previous epidemic and outbreak experiences and the established health policy infrastructure, with a focus on community-level engagement, including contact tracing, “social surveillance”, and volunteerism. Despite the obvious challenges, predictors of success relied heavily on a communal sense of responsibility, playing-field equality, and building social capital through cooperation. Self-governance and first-hand experience with witnessing positive outcomes, such as patient recovery or community successes in reducing incidences, were important in building social capital and contributed to widespread social cohesion [61,73].

As initial shock and disruption are supplanted by a normalizing of pandemic life, community innovation and resilient governance may start to take hold. Despite an overwhelming trend toward the negative, these examples of resilience and adaptation provide a bedrock from which to develop future potential solutions, utilizing local context, social solutions, and effective community engagement and communication.

## 4. Discussion

The reviewed qualitative studies, both ethnographic and descriptive, have demonstrated that responses to the pandemic are at once universal and specific; human emotion unites, while socio-cultural context distinguishes. Social suffering in the face of an overwhelming, disruptive health crisis is expected and unsurprising, as illustrated in research from the profiled LMICs in which fear, uncertainty, stigma, and anxiety pervaded. Structural insufficiency, local understanding, the proliferation of misinformation, and under-resourced health systems all contributed to the destabilization of emotional responses. Resilience was also profiled, underscoring the importance of drilling down to the local context to find solutions, rather than relying on facile “copy–paste” interventions and rigid hierarchical power relations. Nimble, flexible, and local solutions that incorporated traditional explanatory models, and even spiritual technologies, appeared to contribute to resilience, as exemplified in several studies included in this review.

The papers included in this review represent significant variations in geographic and topical areas of inquiry, yet analytically tend toward biomedical reductionism in scope. While explicitly making claims of employing qualitative, phenomenological, and ethnographic methods, many papers made few explicit references to specific scholars or theoretical frameworks for comprehensive analyses and the contextualization of findings within the broad sociological or anthropological literature of epidemics. The use of theory for in-depth analysis in the selected studies was variable, yet some notable exceptions provided rich context and interpretation.

Inayat Ali provided insight into the Pakistani government’s containment of contagion by using Turner’s “social drama” as a way of exploring the multiple crises brought on by COVID-19, in which the public and the state managed tensions in the context of new subjects and ways of being [58]. Researchers in Indonesian Borneo made note of the “illusory truth effect” [128] in their reporting on how misinformation and rumor quickly evolved into accepted truths about COVID-19 through reinforcement and repetition, this having widespread consequences for communities with scant access to reliable information sources [65]. Bhatt, et al. outlined their use of Colaizzi’s phenomenological analysis of interview transcripts for their research in Nepal, yet the findings were reported descriptively by theme and were compared with other empirical research, but were not subjected to theoretical conjecture or analyses [63]. Jones’s comprehensive ethnography in Sierra Leone explored individual meanings and experiences of the COVID-19 crisis, while being contextualized in larger issues of culture, governance, and looking at adaptation and resilience [66]. Similarly, Sumesh et al. incorporated Link and Phelan’s framework for stigma [129] and referenced Merleau-Ponty and other phenomenological theorists in their “pandemic ethnography” in India, yet robust interpretation that engaged with phenomenology was only limited [80]. Tan and Lasco, employing ethnographic investigations into the use of local knowledge around illness concepts, were comprehensive in their interpretations of the political and moral implications of long-held cultural idioms around infection and immunity in the Philippines [81]. Ali’s study of rituals of containment in Pakistan, and Tan and Lasco’s study in the Philippines, both provided the most comprehensive linkages between methodology and interpretation and offered models for ethnographic investigations and theory during this pandemic [59,81]. Østebø et al. provided a comprehensive analysis of the secular–religious debate and embedded their discussion in a deep historical context in Ethiopia, including experiences with previous outbreaks and the cultural forms of “coexisting epistemologies”, as practiced in the Tigray and Amhara communities’ systems of traditional knowledge [72]. Giddens’ concept of “space-time distanciation” was used to analyze the temporal ruptures in everyday life in Burkina Faso, linking socio-cultural context to the passage of time, to distance, and to the influence of the “absent other” [76]. 

While most studies provided insight into community perceptions and the impacts of the COVID-19 pandemic, the framing of analysis and reporting from a dominant biomedical epistemology lacked robustness in considering multiple ontological stances. Field sites in LMICs were predominantly “non-Western”, yet they were approached using knowledge generated from Cartesian biomedicine and epidemiology, often assessing the “correctness” of responses to test knowledge and attitudes. Additionally, studies were mostly “agency-focused” in terms of recommendations going forward, as interventions were mainly at the individual/perceptual level, such as providing communications and education around stigma, but lacking structural or political-economic recommendations. There remain ample opportunities to explore how a rapidly evolving local cultural nosology incorporates COVID-19 into local knowledge structures and the biosocial nature of the pandemic, which was not fully explored in these studies to a satisfactory degree.

Considering the pervasively disruptive nature of the COVID-19 pandemic on the execution of field research in general, few studies explicitly detailed how alterations in data collection were accommodated to comply with pandemic regulations. Bhatt et al., Jones, and Sumesh et al. noted how their data collection methods were adjusted to meet government requirements and to avoid contamination during interviews, including the provision of masks, sanitizers, and maintaining physical distancing and barriers [63,66,80]. Kwaghe et al. described the accommodations for interviewing that were put in place for one participant who was in quarantine isolation at the time of the interview, shifting to a telephone interview rather than conducting it face-to-face, like the other respondents [68]. Amir and Ekoh et al. also referred to abiding by standard COVID-19 regulations in their Methods sections [60,64], while Samuelsen and Toé denoted adjustments in travel and the use of local field assistants during strict lockdowns [76]. Sumesh et al. also included a sub-section on future considerations for changes to interview methods, to accommodate qualitative data collection during pandemic conditions [80]. However, most studies described their data collection methods as per usual, with little direct mention of COVID-19-induced adjustments to the status quo in their research methodologies.

Despite rich opportunities for theoretical exploration using qualitative data from empirical studies during COVID-19, the dearth of analytical depth and the decision to opt for thematic description and comparison demonstrates an unmet need for further research. Although most of the researchers were academics from the same country or cultural areas being studied, the majority hailed from positivist clinical sciences, and Enlightenment intellectual traditions. Qualitative analysis tended toward reductive description rather than a nuanced and significant contribution to the social or anthropological theory of epidemics. This was especially apparent in the general lack of robust analytics incorporating both macro (historical, political-economic) and micro (individual, psychological, economic) influences and impacts within the socio-cultural milieux under investigation.

Additionally, there was scant evidence of critical analysis framing the studies in broader questions of the power and legitimation of pandemic knowledge in the unfolding novel crisis. Emphasis was heavily placed on the hegemonic biomedical paradigm of knowledge production, used as the template by which to grade responses and the performance of interventions. Little is reported from an anthropological perspective of multiple ontologies, or disease framing via local explanatory models and illness narratives, which underscores the predominance of the top-down colonial model of Western scientific knowledge paradigms in pandemic responses, with the notable exceptions of Østebø et al. in Ethiopia, Tan and Lasco in the Philippines, Ali in Pakistan, Jones in Sierra Leone, and Sumesh et al. in India [58,66,72,80,81].

## 5. Conclusions

As illustrated by the reviewed studies, local context and community engagement were indispensable for designing public health interventions to meet the challenges of the pandemic. The complex dynamic of perception, experience, and behavior arising from the entanglement of individuals, communities, and intangibles of structure, differing notions, and understanding of pandemic meanings was brought into sharper relief through ethnographic and qualitative inquiry. The synthesis and analysis of social research outlining local context and pandemic responses is crucial for developing appropriate policy and engagement toward ending the crisis.

Inter- and intrastate politics, influenced by economics, contemporary history and political instability, religion, and structural influences creating inequity and marginalization are major factors determining pandemic responses in LMICs. Individual experiences in the communities reflected structural inequity, where sense-making at the local level was a culmination of these proximate and distal forces, one that usually resulted in negative health outcomes, stigma, social rejection, and so on. While several exemplars of community resilience were identified, social ruptures caused by the COVID-19 pandemic necessitate large-scale rectification, requiring improved commitments by the State and civil society toward future prevention and response. This review also underscores the importance of facilitating and improving public–private cooperation and policy, as most of these studies reveal disjunctures between civil society, the public at large, and the governing bodies responsible for wielding legislative, implementation, and enforcement powers to contain the pandemic.

## Figures and Tables

**Figure 1 ijerph-18-12063-f001:**
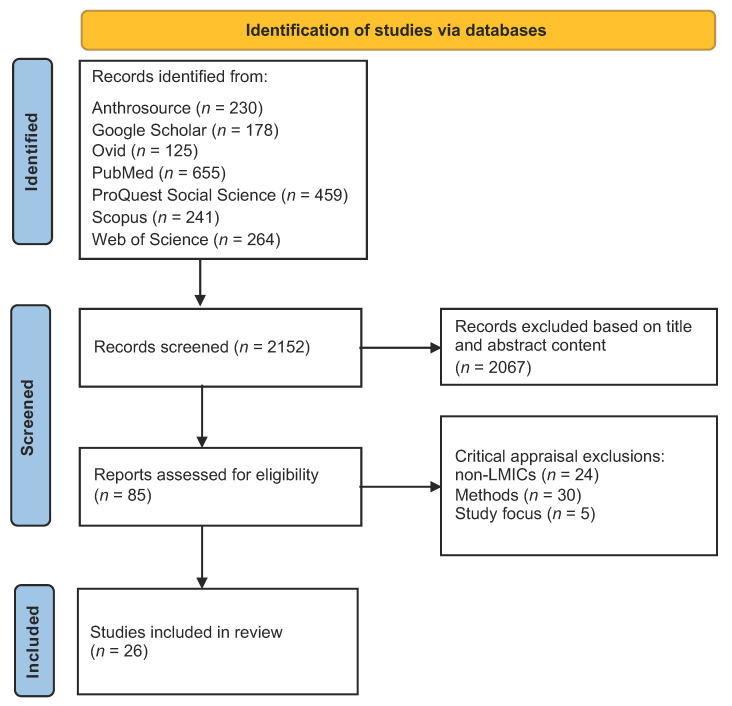
PRISMA flow diagram of the study selection process.

**Figure 2 ijerph-18-12063-f002:**
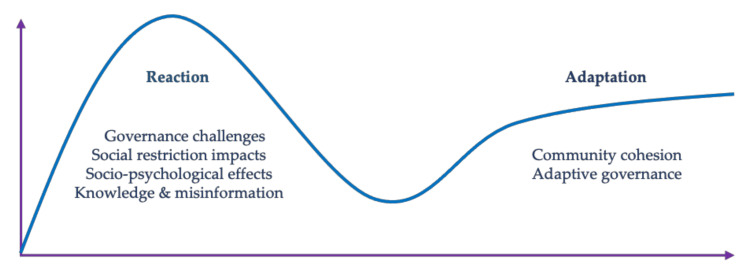
Themes categorizing studies from the literature review focused on reaction and adaptation.

**Table 1 ijerph-18-12063-t001:** Inclusion and exclusion criteria.

Topic	Inclusion Criteria (Met All)	Exclusion Criteria (Met Any)
Scope	Focus on COVID-19 impacts using social research methods Primary qualitative data collection from community or participatory settings Research conducted in LMICs	Studies conducted in non-LMICs Quantitative methodology Emphasis on virtual, digital, or distance data collection, such as phone or teleconferencing interviews or online surveys Methodologically low rigor
Type	Peer-reviewed journal articles publishing data from empirical studies	Grey literature, systematic reviews, published protocols, or commentaries
Language	English terms used for database search	Non-English articles
Timeline	Published after December 2019 through August 2021	Data collected prior to December 2019

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
