# Peer review of "Community-Level Experiences, Understandings, and Responses to COVID-19 in Low- and Middle-Income Countries: A Systematic Review of Qualitative and Ethnographic Studies"

_ijerph, 2021, doi:10.3390/ijerph182212063_

Round 1

Reviewer 1 Report

Comments

It is an investigation based on qualitative, ethnographic and descriptive studies, therefore, with a high degree of subjectivity. However, the authors managed to obtain an important contribution, from several angles, on the influence of COVID-19 on LMICs, but in an incomplete way.
The gap lies in the lack of consideration in the study of the countries of South America and the Caribbean, as many or almost all of these countries are considered by the World Bank to be developing countries with low national income. They are also countries of the "Global South" (line 55). Unless they had included studies that were not processed in the database considered by the authors. 
It remains, however, the warning for future developments of this theme.

No other region in the world has suffered more from the pandemic than the South American region, in terms of GDP and in terms of mortality rate (about 2.1 m of people), when the unemployment rate was 14%.

Suggestions

Just start by exploring the following World Bank Group link: https://www.worldbank.org/en/news/factsheet/2020/04/02/world-bank-response-to-covid-19-coronavirus-latin-america-and-caribbean.

At least some of South American countries should be included in Table 2, even with some lack of representation of studies included in the literature.
Therefore, Table 2 is imprecise, as I said before. In these countries, like those covered by the systematic review, police violence was also a reference in the behaviour of political authorities, being theoretically democratic countries in a region of the world with one of the highest levels of corruption.

On the other hand, the social impacts of the COVID-19 pandemic in terms of lockdown, physical distance, the use of masks, hand washing, working hours and restrictions on mobility (lines 434-435) were brakes used in every country in the world. Poverty and inequality have increased more in the southern hemisphere (and beyond). 

The political governance of the crisis in the countries studied should have been more explored or, at least, more highlighted, but here is the alert for future works, more in the nature of Political Science. In fact, indirectly, the authors affirm (lines 593-595) but have not developed.

Also for future work, the concentration of research must now be focused on public and private policies to recover from the catastrophic situation these countries have reached, even if the future results have a degree of uncertainty and unreliable evidence.

About spirituality, it will also be difficult to understand how a minority wants to impose its religious ethics on an entire society. Those who have faith have the right to affirm what they believe and live accordingly. Furthermore, citizenship also gives you legitimacy to seek to influence others in the sense of your beliefs and lifestyle, but nobody no longer have any right to want to impose your religious ethics by force or law on those who do not have faith. This is abusive. There is no other way to say it.

This is more analysis for future developments on this theme.

Author Response

PLEASE SEE ATTACHED REVISED MANUSCRIPT

Manuscript ID:

ijerph-1429696

Authors’ Responses to Reviewer 1 Comments

Point 1: It is an investigation based on qualitative, ethnographic and descriptive studies, therefore, with a high degree of subjectivity. However, the authors managed to obtain an important contribution, from several angles, on the influence of COVID-19 on LMICs, but in an incomplete way.

The gap lies in the lack of consideration in the study of the countries of South America and the Caribbean, as many or almost all of these countries are considered by the World Bank to be developing countries with low national income. They are also countries of the "Global South" (line 55). Unless they had included studies that were not processed in the database considered by the authors. It remains, however, the warning for future developments of this theme.

No other region in the world has suffered more from the pandemic than the South American region, in terms of GDP and in terms of mortality rate (about 2.1 m of people), when the unemployment rate was 14%.

Suggestions

Just start by exploring the following World Bank Group link: https://www.worldbank.org/en/news/factsheet/2020/04/02/world-bank-response-to-covid-19-coronavirus-latin-america-and-caribbean.

At least some of South American countries should be included in Table 2, even with some lack of representation of studies included in the literature.

Therefore, Table 2 is imprecise, as I said before. In these countries, like those covered by the systematic review, police violence was also a reference in the behaviour of political authorities, being theoretically democratic countries in a region of the world with one of the highest levels of corruption.

On the other hand, the social impacts of the COVID-19 pandemic in terms of lockdown, physical distance, the use of masks, hand washing, working hours and restrictions on mobility (lines 434-435) were brakes used in every country in the world. Poverty and inequality have increased more in the southern hemisphere (and beyond).

Response 1:

We strongly agree that Latin America has been detrimentally and dramatically affected by COVID-19. Latin American countries were not excluded from this review. We acknowledge the importance of this region and topic area. It is likely that in the initial database searches A) any ethnographies from Latin America identified in the initial database search were excluded due to publication language (Spanish, Portuguese, etc were not included), B) papers were excluded based on critical appraisals or C) the databases didn’t pick up the papers in the initial search.

Referencing the review’s inclusion & exclusion criteria (table 1 at line 144 in revised manuscript):

Topic

Inclusion Criteria (met all)

Exclusion Criteria (met any)

Scope

-Focus on COVID-19 impacts using social research methods

-Primary qualitative data collection from community or participatory settings

-Research conducted in LMICs

-Studies conducted in non-LMICs

-Quantitative methodology

-Emphasis on virtual, digital, or distance data collection such as phone or teleconferencing interviews or online surveys

-Methodologically low rigor

Type

Peer-reviewed journal articles publishing data from empirical studies

Grey literature, systematic reviews, published protocols, or commentaries

Language

English terms used for database search

Non-English articles

Timeline

Published after Dec 2019 through Aug 2021

Data collected prior to Dec 2019

and using the World Bank definitions for LICs, LMICs, UMICs, UICs, only the following six countries were eligible for inclusion as LMICs in Latin America:

Belize, Bolivia, El Salvador, Haiti, Honduras, Nicaragua

There were no LICs in Latin America, and the remaining Latin American countries are classified as UMICs or UICs. This data was taken from the World Bank:

https://datahelpdesk.worldbank.org/knowledgebase/articles/906519-world-bank-country-and-lending-groups

https://datatopics.worldbank.org/world-development-indicators/the-world-by-income-and-region.html

https://blogs.worldbank.org/opendata/new-world-bank-country-classifications-income-level-2021-2022

In the initial systematic search of databases, no papers from these six countries were identified that were eligible based on the general search terms employed. Upon further and directed searches using country specific keywords with these six counties, one additional paper  was identified for potential inclusion which met all inclusion/exclusion criteria, from Bolivia entitled "The impact of COVID-19 pandemic on frail health systems of low- and middle-income countries: The case of epilepsy in the rural areas of the Bolivian Chaco". This paper was added to Table 2 and included in the revised manuscript text. 

Point 2: The political governance of the crisis in the countries studied should have been more explored or, at least, more highlighted, but here is the alert for future works, more in the nature of Political Science. In fact, indirectly, the authors affirm (lines 593-595) but have not developed.

Response 2:

We agree and have amended the narrative text in response (see below).

Point 3: Also for future work, the concentration of research must now be focused on public and private policies to recover from the catastrophic situation these countries have reached, even if the future results have a degree of uncertainty and unreliable evidence.

Response 3:

We agree and have amended the conclusion to refer to this issue.

Point 4: About spirituality, it will also be difficult to understand how a minority wants to impose its religious ethics on an entire society. Those who have faith have the right to affirm what they believe and live accordingly. Furthermore, citizenship also gives you legitimacy to seek to influence others in the sense of your beliefs and lifestyle, but nobody no longer have any right to want to impose your religious ethics by force or law on those who do not have faith. This is abusive. There is no other way to say it.

Response 4:

Agreed, the conflation of religious authority and the governance of the pandemic was a difficult issue encountered by many of these studies. We have reported results in line with what the study authors have concluded, which is that religion is an important component of consideration for all of these communities, as religious beliefs and practices strongly influenced individual perceptions, experiences, and pandemic responses.

Manuscript revisions based on Reviewer comments:

Line 54: footnote added to clarify definition of LMICs based on World Bank classifications

Line 151 added to clarify databases used

Figure 1, line 194, PRISMA diagram revised to include additional Nicoletti et al. study

Table 2, line 230, revised to include additional Nicoletti et al. study

Lines 602-613 added for additional consideration as per Point 2 above

Lines 853-865 added to conclusion

Reviewer 2 Report

Based on the analysis of 25 selected researches, this study discusses various situations during COVID-19 pandemic in communities of several low and medium-income countries such as India, and reveals the significance of local context and community engagement for countries to meet the challenge of pandemic. Here are a few questions I would like to debate with the authors:

  1. The authors need to specify the criteria to define "low and medium-income countries". If they are World Bank standards, the researches from China should be taken into account.
  2. Dose some certain 25 countries such as Indonesia sufficiently represent for "low and medium-income countries"? Perhaps the authors could reconsider the title and related expressions.
  3. Although the disabilities account for approximately 15% in the world’s population, it has not been mentioned in the public health policy responding to COVID-19 in many countries (including Australia). As far as I know, some valuable researches have paid closely attention to the disabled in the pandemic. Considering that most of the disabled people live in developing countries, I think these references should be included in this research. So if possible, authors could analyze issues from the disability perspectives in their research. If objective conditions do not allow, additional paragraph about this limitation should be mentioned.

Author Response

PLEASE SEE ATTACHED REVISED MANUSCRIPT

Manuscript ID:

ijerph-1429696

Authors’ Responses to Reviewer 2 Comments

Based on the analysis of 25 selected researches, this study discusses various situations during COVID-19 pandemic in communities of several low and medium-income countries such as India, and reveals the significance of local context and community engagement for countries to meet the challenge of pandemic. Here are a few questions I would like to debate with the authors:

Point 1: The authors need to specify the criteria to define "low and medium-income countries". If they are World Bank standards, the researches from China should be taken into account.

Response 1:

We strongly agree that China is an extremely important country that has been affected by COVID-19, however they were excluded as they are classified by the World Bank as an Upper Middle-Income Country. Decisions on countries to include in this systematic review of the literature were made in reference to the inclusion & exclusion criteria (table 1 at line 141): 

Topic

Inclusion Criteria (met all)

Exclusion Criteria (met any)

Scope

-Focus on COVID-19 impacts using social research methods

-Primary qualitative data collection from community or participatory settings

-Research conducted in LMICs

-Studies conducted in non-LMICs

-Quantitative methodology

-Emphasis on virtual, digital, or distance data collection such as phone or teleconferencing interviews or online surveys

-Methodologically low rigor

Type

Peer-reviewed journal articles publishing data from empirical studies

Grey literature, systematic reviews, published protocols, or commentaries

Language

English terms used for database search

Non-English articles

Timeline

Published after Dec 2019 through Aug 2021

Data collected prior to Dec 2019

and using the World Bank definitions for LICs, LMICs, UMICs, and UICs, taken from the World Bank databases found here:

https://datahelpdesk.worldbank.org/knowledgebase/articles/906519-world-bank-country-and-lending-groups

https://datatopics.worldbank.org/world-development-indicators/the-world-by-income-and-region.html

https://blogs.worldbank.org/opendata/new-world-bank-country-classifications-income-level-2021-2022

China’s GNI is currently $10,610 USD, placing it in the UMIC bracket, please kindly refer to:

https://data.worldbank.org/indicator/NY.GNP.PCAP.CD?locations=CN

Point 2: Dose some certain 25 countries such as Indonesia sufficiently represent for "low and medium-income countries"? Perhaps the authors could reconsider the title and related expressions.

Response 2:

Similar to point 1, the Low- and Middle-Income Country (LMIC ) classification of countries was taken from World Bank definitions (see above). Based on this classification, and in consideration of additional inclusion and exclusion criteria for the systematic review, we identified ethnographic studies which were performed in communities in the following countries (highlighted in the table below taken from

https://datahelpdesk.worldbank.org/knowledgebase/articles/906519-world-bank-country-and-lending-groups):

LOW-INCOME ECONOMIES ($1,045 USD OR LESS)     

Afghanistan

Guinea-Bissau

Somalia

Burkina Faso

Korea, Dem. People's Rep  

South Sudan

Burundi

Liberia

Sudan

Central African Republic

Madagascar

Syrian Arab Republic

Chad

Malawi

Togo

Congo, Dem. Rep

Mali

Uganda

Eritrea

Mozambique

Yemen, Rep.

Ethiopia

Niger

Gambia, The

Rwanda

Guinea

Sierra Leone

LOWER-MIDDLE INCOME ECONOMIES ($1,046 TO $4,095 USD)

Angola

Honduras

Philippines

Algeria

India

Samoa

Bangladesh

Indonesia

São Tomé and Principe

Belize

Iran, Islamic Rep

Senegal

Benin

Kenya

Solomon Islands  

Bhutan

Kiribati

Sri Lanka

Bolivia

Kyrgyz Republic

Tanzania

Cabo Verde

Lao PDR

Tajikistan

Cambodia

Lesotho

Timor-Leste

Cameroon

Mauritania

Tunisia

Comoros

Micronesia, Fed. Sts.

Ukraine

Congo, Rep.

Mongolia

Uzbekistan

Côte d'Ivoire

Morocco

Vanuatu

Djibouti

Myanmar

Vietnam

Egypt, Arab Rep.

Nepal

West Bank and Gaza

El Salvador

Nicaragua

Zambia

Eswatini

Nigeria

Zimbabwe

Ghana

Pakistan

Haiti

Papua New Guinea

Research papers from UMICs and UICs were excluded from this review.

Point 3: Although the disabilities account for approximately 15% in the world’s population, it has not been mentioned in the public health policy responding to COVID-19 in many countries (including Australia). As far as I know, some valuable researches have paid closely attention to the disabled in the pandemic. Considering that most of the disabled people live in developing countries, I think these references should be included in this research. So if possible, authors could analyze issues from the disability perspectives in their research. If objective conditions do not allow, additional paragraph about this limitation should be mentioned.

Response 3:

Disabilities and their impacts on communities during the COVID-19 are important aspects of how people responded in developing countries. Any topics that authors identified and studied in their ethnographic accounts were included in the systematic review. None of the included ethnographic papers which met inclusion and exclusion criteria focused on or mentioned disability in their studies. This should not detract from its importance as an issue, but we were unable to expand on this issue as it was beyond the scope of the identified papers included in this review.

One additional paper has been identified and included in this review which met the inclusion/exclusion criteria from Bolivia, and which focuses on the case study of epilepsy during the pandemic: The impact of COVID-19 pandemic on frail health systems of low- and middle-income countries: The case of epilepsy in the rural areas of the Bolivian Chaco. The tables and narrative text have been revised accordingly to accommodate this late addition.

Manuscript revisions based on Reviewer comments:

Line 54: footnote added to clarify definition of LMICs based on World Bank classifications

Line 73 added to include mention of disability

Line 151 added to clarify databases used

Figure 1, line 194, PRISMA diagram revised to include additional Nicoletti et al. study

Table 2, line 230, revised to include additional Nicoletti et al. study

Line 502-513 revised to include Nicoletti et al. study

Lines 602-613 added for additional consideration as per Point 2 above

Lines 853-865 added to conclusion

Reviewer 3 Report

While the topic is very interesting, and the paper per whole is a very nice readding, I mult mention some missing facts from a qualitative research, as the authors claim this is what they used as method.

1) What are the research questions? A qualitative research is structured around a set of research questions that would be answered through the research conducted.

2) The results and discussions should try and answer those questions.

The Conclusions sections should be enhaced.

Author Response

Manuscript ID:

ijerph-1429696

Authors’ Responses to Reviewer 3 Comments

Point 1: While the topic is very interesting, and the paper per whole is a very nice readding, I must mention some missing facts from qualitative research, as the authors claim this is what they used as method.

1) What are the research questions? A qualitative research is structured around a set of research questions that would be answered through the research conducted.

Response 1:

The research question guiding this systematic review is found at Line 101:

“The fundamental question for this review is, what do we know about the meaning of COVID-19 in the communities it is most affecting, and how is this represented in the empirical literature?”

We have amended as follows:

Line 101: “The fundamental research question for this review is, what do we know about the meaning of COVID-19 in the communities it is most affecting, and how is this represented in the empirical literature?”

Point 2: The results and discussions should try and answer those questions.

Response 2:

Based on the above response, the results and discussions seem to adequately answer the research question, as the review was designed with this aim.

Point 3: The Conclusions sections should be enhaced.

Response 3:

We agree and the conclusions have been amended, see below.

Manuscript revisions based on Reviewer comments:

Line 54: footnote added to clarify definition of LMICs based on World Bank classifications

Line 73 added to include mention of disability

Line 151 added to clarify databases used

Figure 1, line 194, PRISMA diagram revised to include additional Nicoletti et al. study

Table 2, line 230, revised to include additional Nicoletti et al. study

Line 502-513 revised to include Nicoletti et al. study

Lines 602-613 added for additional consideration as per Point 2 above

Lines 853-865 added to conclusion

Round 2

Reviewer 3 Report

I recommend a better differentiation of the research question from the rest of the text, as it may get lost in the reading.

Otherwise, I agree with your additions and it is overall a very intersting paper.